# Immunomodulatory Effects of Chicken Broth and Histidine Dipeptides on the Cyclophosphamide-Induced Immunosuppression Mouse Model

**DOI:** 10.3390/nu14214491

**Published:** 2022-10-25

**Authors:** Jian Zhang, Xixi Wang, He Li, Cunshe Chen, Xinqi Liu

**Affiliations:** 1Beijing Advanced Innovation Center for Food Nutrition and Human Health, School of Food and Health, Beijing Technology and Business University, Beijing 100048, China; 2Beijing Advanced Innovation Center for Food Nutrition and Human Health, Department of Nutrition and Health, China Agricultural University, Beijing 100193, China; 3China Animal Disease Control Center, Beijing 102618, China

**Keywords:** chicken broth hydrolysate, histidine dipeptides, carnosine, anserine, immunomodulatory, cyclophosphamide, immunosuppressed mice

## Abstract

The carnosine and anserine, which represent histidine dipeptides (HD), are abundant in chicken broth (CB). HD are endogenous dipeptide that has excellent antioxidant and immunomodulatory effects. The immunomodulatory effect of CB hydrolysate (CBH) and HD in cyclophosphamide (CTX)-induced immunosuppressed mice was examined in this study. CBH and HD were given to mice via oral gavage for 15 days, accompanied by intraperitoneal CTX administration to induce immunosuppression. CBH and HD treatment were observed to reduce immune organ atrophy (*p* < 0.05) and stimulate the proliferation of splenic lymphocytes (*p* < 0.05) while improving white blood cell, immunoglobulin M (IgM), IgG, and IgA levels (*p* < 0.05). Moreover, CBH and HD strongly stimulated interleukin-2 (IL-2) and interferon-gamma (IFN-γ) production by up-regulating IL-2 and IFN-γ mRNA expression (*p* < 0.05) while inhibiting interleukin-10 (IL-10) overproduction and IL-10 mRNA expression (*p* < 0.05). In addition, CBH and HD prevented the inhibition of the nitric oxide (NP)/cyclic guanosine monophosphate-cyclic adenosine monophosphate (cGMP-cAMP)/protein kinase A (PKA) signaling pathway (*p* < 0.05). These results indicate that CBH and HD have the potential to prevent immunosuppression induced by CTX. Our data demonstrate that CBH can effectively improve the immune capacity of immunosuppressed mice similar to the same amount of purified HD, which indicates that CBH plays its role through its own HD.

## 1. Introduction

Chicken broth (CB) is an ideal food for those recovering from illness since it is nutritious, easily digestible, and contains high levels of proteins, free amino acids, and polysaccharides [1,2]. Since it is easy to prepare, healthy, and delicious, CB has become a typical dish on family and restaurant tables. In Asia, CB has long been used as a nutritional supplement because it boosts immunity, combats fatigue, and prevents colds [3].

Histidine dipeptides (HD) represent a class of water-soluble dipeptides typically found in the skeletal muscles and brain tissue of many vertebrates and primarily include carnosine (CAR), anserine (ANS), and balenine [4]. The CAR and ANS structures are shown in Figure 1A. CAR and ANS can be synthesized in the body or ingested through diet. CAR and its derivatives are present in substantial levels in red and white meats (beef, chicken, and pork), as well as fish, in the human diet [5]. The order of the HD content in various common animals are as follows: Turkey > chicken > horse > pig > rabbit > cattle [6]. The HD content of meat products can be altered via Specific food processing technology. Papain and flavourzyme protease are often used in food processing to improve CB flavor and protein concentration, and the subsequent CB exhibited better taste and flavor than traditional chicken soup [7]. In addition, chicken meat extract contains high CAR and ANS concentrations at a 1:2 to 1:3 ratio [8].

HD can fulfill many physiological functions in the human body. The antioxidant and anti-inflammatory characteristics of HD are responsible for the majority of its beneficial effects, including immunological modulation, anti-aging, anti-neurodegenerative illness, anti-diabetes, and others. The ability of HD to act as an antioxidant is its their primary function [9]. CAR supplementation can decrease levels of advanced glycation end-products, malondialdehyde, protein carbonyl, and advanced oxidized protein product, as well as the generation of reactive oxygen species in the serum and liver of elderly rats [10]. Antonini et al. [11] proved that a diet rich in meat and CAR increased the antioxidant activity of human serum. Additionally, CAR has been shown to play an anti-aging role by suppressing telomere shortening, antioxidant activity, carbonyl scavenging, glycolysis suppression, the upregulation of mitochondrial activity, and the rejuvenation of senescent cells [12]. It also improves collagen content in the skin and may prevent skin aging [13]. Daily ANS/CAR supplementation is beneficial to the memory, cognitive function, and physical activity of elderly people [14]. In addition, CAR was indicated to possess immune regulation effects [15]. During the last 5 years, the ability of CAR to modulate different activities of immune cells such as macrophages has been demonstrated [16,17]. It has been demonstrated that the interaction of CAR with particular receptors on the cell membrane could modify macrophage function by enhancing their phagocytotic activity [18,19]. It has also been demonstrated that HD can influence NO production and macrophage polarization [20,21]. CAR maintained spleen lymphocyte number by inhibiting lymphocyte apoptosis and stimulating lymphocyte proliferation, thus preventing immunocompromise in mice [22]. Deng et al. [23] suggested that CAR can protect murine bone marrow cells from cyclophosphamide (CTX)-induced DNA damage via its antioxidant activity. Thus, HD has a significant impact on immune response regulation.

HD is a beneficial functional component of CB. The resistance immunosuppression function of CB may be played through HD. Therefore, we increased the HD content of CB by enzymatic hydrolysis pretreatment with suitable enzymes in this study. Additionally, in order to understand the immunity-enhancing effect of CB hydrolysate (CBH) and identify its major functional constituents, we studied the immunomodulatory effects of CBH and HD on CTX-treated mice. If CBH aids in the recovery of immunosuppression, it could be a safe and effective booster for postoperative patients.

## 2. Materials and Methods

### 2.1. Materials and Chemicals

Hetian chicken was purchased from the Hetian Chicken Development Co., Ltd. (Fujian, China). Neutrase, Bromelain, Papain, Flavourzyme, and Protamex were obtained from the Shanghai Yuanye Bio-Technology Co., Ltd. (Shanghai, China). CTX was purchased from the Jiangsu Hengrui Medicine Co., Ltd. (Jiangsu, China). The CAR and ANS standards were acquired from the Shanghai Yuanye Bio-Technology Co., Ltd. (Shanghai, China). Levamisole was purchased from the Novozymes (China) Investment Co., Ltd. (Beijing, China). Roswell Park Memorial Institute (RPMI) 1640 medium and fetal bovine serum (FBS) were procured from Gibco (Grand Island, NY, USA). Concanavalin A (con A) and 3[4,5-dimethylththiazoyl-2-yl]2,5-diphenyltetrazolium bromide (MTT) were purchased from Sigma Aldrich Co. (St Louis, MO, USA). The enzyme-linked immunosorbent assay (ELISA) kits for the immunoglobulin M (IgM), immunoglobulin G (IgG), immunoglobulin A (IgA), interleukin-2 (IL-2), interleukin-10 (IL-10), interferon-gamma (IFN-γ), nitric oxide (NO), cyclic guanosine monophosphate (cGMP), cyclic adenosine monophosphate (cAMP), and protein kinase A (PKA) were purchased from the Meimian Biotechnology Co. Ltd. (Jiangsu, China). All chemical reagents were of analytical grade.

### 2.2. Preparation of the CB

#### 2.2.1. Extracting the HD from the Chicken Breast

A meat grinder was used to mince the chopped chicken breast meat. Following that, 100 g of the minced chicken breast flesh was combined with 100 mL of deionized water and maintained in a water bath at 100 °C for 15 min to inactivate the CAR enzymes, followed by 30 min of ultrasonic extraction. Next, the mixture was centrifuged at 12,000 r/min for 8 min at 4 °C. The supernatant was then filtered to a constant volume of 1000 mL with a 0.22 μm nylon filter.

#### 2.2.2. Traditional CB

The chicken breast was cut into small pieces of 1 cm^3^. Next, 200 mL deionized water was added to 100 g minced chicken breast flesh. The mixture was added to a steamer and stewed for 4 h at a pressure of 0.07 MPa at 100 °C. The prepared CB was centrifuged at 4 °C for 8 min at 12,000 r/min. The supernatant was then filtered to a constant volume of 1000 mL with a 0.22 μm nylon filter.

#### 2.2.3. Enzymolysis of the CB

The chicken breast was cut into small pieces of 1 cm^3^. Subsequently, 200 mL deionized water was added to 100 g minced chicken breast flesh. The mixture was hydrolyzed using Neutrase (pH 7, 45 °C), Papain (pH 6.5, 55 °C), Bromelain (pH 6.5, 45 °C), Flavourzyme (pH 7.5, 50 °C), and Protamex (Papain: Flavourzyme 1:1, pH 7, 45 °C) according to the amount of enzyme added at 1000 U/g, respectively, adjusted to the optimum temperature and pH of the enzyme. In the process of enzymatic hydrolysis, 1 mol/L NaOH was added to adjust the pH value of the system to remain constant, and the system was enzymatic for 2 h. After enzymatic hydrolysis, the mixture was heated to 100 °C for 10 min to inactivate the enzymes. Next, the mixture was placed in a steamer and stewed for 4 h at a pressure of 0.07 MPa at 100 °C. The CBH was cooled and centrifuged at 4 °C for 30 min at 12,000 r/min. The supernatant was then filtered to a constant volume of 1000 mL with a 0.22 μm nylon filter.

### 2.3. Protein and Free Amino Acid Analysis of the CB

The Kjeldahl method (Kjeltec 8000, FOSS Analytical A/S, Denmark) was used to assess the protein concentrations in the CB. The CB was centrifuged for 10 min at 9600 r/min under 4 °C. The fat from the supernatant was removed and lyophilized to produce CB Powder. Separately, CB Powder was redissolved by Sodium Loading Buffer. The solutions were analyzed with an automatic amino acid analyzer (Biochrom 30+, Biochrom Ltd., Cambridge, England) after filtration through a 0.45 μm nylon filter membrane (Cleman, Beijing, China). Absorbance was recorded at 570 nm and 440 nm.

### 2.4. UPLC Analysis of the HD

UPLC was used to analyze the CAR and ANS in chicken breast, CB, and CBH. A 10 μL sample was injected into the UPLC system (Agilent, 1290 Infinity II), and the separation was performed using the Poroshell 120 HILIC-Z column (2.1 mm × 100 mm, 2.7 μm). The mobile phase was composed of 90% acetonitrile and 0.1% trifluoroacetic acid at a liquid flow rate of 0.2 mL/min. The detection wavelength was 210 nm.

### 2.5. Animals

In this case, 50 eight-week-old ICR male mice (SPF Beijing Biotechnology Co., Ltd., Beijing, China) weighing 35 ± 2 g were used. All mice were housed under controlled temperature (22 ± 2 °C) and humidity (50–60%), with a 12 h light: 12 h dark cycle. The experimental protocol was approved by the Institutional Animal Care and Use Committee at the Pony Testing International Group Co., Ltd., Beijing, China (No. PONY-2021-FL-03).

### 2.6. Experimental Procedures

After a one-week acclimatization period, all mice were randomly divided into five groups (10 per group): (1) Normal group, (2) CTX group, (3) Levamisole group, (4) CBH group, and (5) HD group. Mice in the Levamisole group received Levamisole daily by oral gavage for 15 days at a dose of 10 mg/kg body weight. Mice in the CBH group and HD group received CBH and HD daily by oral gavage for 15 days. After negative pressure concentration, the HD concentration in the CBH was 30 mg/mL, including 9.41 mg/mL CAR and 20.59 mg/mL ANS. The HD doses were equivalent to the HD concentration and proportion in the CBH. Whereas mice in the Normal group and CTX group were treated with the same volume of distilled water. Each group of mice had a gavage volume of 0.1 mL (10 g·bw)^−1^. The immunosuppressed mouse model was established by intraperitoneal injection of CTX. On days 13 to 15, the mice in groups (2) to (5) were intraperitoneally administered with CTX (80 mg/kg/d). The identical volume of physiological saline was given intraperitoneally to group (1). The mice were fasted for 12 h after the last injection and then executed by cervical dislocation.

### 2.7. Determination of the Body Weight and Immune Organ Index

Take measurements of the mice’s body weight before and after the experiment. After the mice were sacrificed, the thymus and spleen tissues were immediately dissected, washed in pre-cooled normal saline at 4 °C, dried using filter paper, and weighed. The organ index was calculated as follows:Organ index (mg/g) = Organ weight (mg)/Bodyweight (g)(1)

### 2.8. Determination of Splenic Lymphocytes Proliferation

The spleens of the sacrificed mice were aseptically dissected and placed in cold Hank’s solution. The spleens were then pulverized using a sterilized glass rod and gently pressed through a 200 mesh, sterile metal strainer, after which the single-cell suspension was obtained. The suspension was centrifuged at 1000 r/min for 10 min at 4 °C after being rinsed twice with Hank’s solution. The cells were then moved to an RPMI-1640 medium with 10% FBS after the supernatant fluid was discarded. The cell concentration was adjusted to 3 × 10^6^ cells/mL. The cells were seeded at a density of 3 × 10^6^ cells per well in a 24-well flat-bottomed plate with or without Con A (7.5 µg/mL) and incubated with 5% CO_2_ for 72 h at 37 °C. The plates were then centrifuged at 200× *g* for 10 min, after which the splenocyte culture supernatants were collected and stored at −80 °C until analysis for cytokines. Next, the splenic lymphocyte proliferation was detected by MTT assay.

### 2.9. Hematological Analyses

Blood was collected from the eyes of the mice 12 h after the last drug administration. The white blood cells (WBC), neutrophils (NEU), lymphocytes (LYM), red blood cells (RBC), hemoglobin (HGB) concentration, and platelet (PLT) number were determined using a hematology analyzer (HEMAVET 950, Drew Scientific Group, Dallas, TX, USA).

### 2.10. Assay of Immunoglobulins in the Serum

Serum was extracted from the blood taken from the eyes of mice after centrifugation. The serum IgM, IgG, and IgA concentrations were determined using ELISA kits.

### 2.11. Measurement of the Cytokines

The IL-2, IL-10, IFN-γ, and NO levels in the splenocyte culture supernatants were detected using ELISA Kits.

### 2.12. Quantitative Real-Time Polymerase Chain Reaction (qRT-PCR) Analysis

In this case, qRT-PCR was used to evaluate the mRNA expression levels of IL-2, IL-10, and IFN-γ. The mice’s spleens were extracted in a sterile setting and washed twice in PBS. The spleens’ total RNA was isolated using Trizol reagent. The RNA concentration and purity were determined using an Ultra-micro spectrophotometer (Nanodrop 2000c, Thermo Fisher Scientific, Chicago, IL, USA). The total RNA (2 μg) was converted into cDNA using PrimeScript RT Master Mix (Takara-bio, Shiga, Japan), while qRT-PCR amplification was performed using Probe qPCR Mix (Takara-bio, Japan). The 18s rRNA was used to normalize the IL-2, IL-10, and IFN-γ mRNA expression. The relative expression levels of the target genes were calculated based on 2^−ΔΔCt^.

### 2.13. Measurement of cAMP/cGMP Levels and PKA Activity

After an incubation period of 72 h, the splenocytes were rinsed three times with PBS and centrifuged at 1500 r/min for 5 min at 4 °C. RIPA buffer was used to lyse the cells for 30 min. The splenocyte supernatants were collected via centrifugation at 12,000 rpm for 5 min at 4 °C. The cAMP/cGMP levels and PKA activity were detected using ELISA kits.

### 2.14. Statistical Analysis

Data were expressed as the mean ± standard deviation (SD). The results were analyzed with one-way analysis of variance (ANOVA) followed by Tukey’s method, using SPSS 23 software, while the graphs were created using Graph Pad Prism 7.

## 3. Results and Discussion

### 3.1. Characterization of the CBH

As shown in Figure 1B, CB had a protein content of 3.59 ± 0.07 g/100 mL. CB contained 334.43 mg/100 mL of total free amino acids, with His being the most abundant, followed by Glu, Ala, and Ser. In this study, chicken breasts containing a high level of HD were selected to prepare the CB. Enzymatic hydrolysis is a standard method used during chicken processing to improve the flavor and taste of the products [7]. It also increases the hydrolytic degree of chicken protein and produces more peptides that are easily absorbed. Different proteases were used in the experiment to prepare the CBH. The results showed that different enzymatic hydrolysis treatments significantly increased the HD content in the CBH. As shown in Figure 1B, ANS content in the CBH ranked as follows: Protamex > Papain > Flavourzyme > Bromelain > Neutrase. The ANS content in the CB hydrolyzed with Protamex was the highest and significantly exceeded that in the other four enzymolytic CB groups (*p* < 0.05). Furthermore, CAR content was considerably greater in the CB hydrolyzed with Papain and Protamex than in the other three enzymolytic CB groups (*p* < 0.05). Therefore, after enzymatic hydrolysis using Protamex, the HD content of the CBH can be effectively increased during the cooking process.

### 3.2. The Effect of CBH on the Body Weight and Immune Organ Index

Bodyweight was a good predictor of the mice’s growth status. The mice were randomly assigned to five groups based on body weight after a week of acclimatization using the randomized design method. Table 1 showed that although no significant differences were evident between the final weight of the mice from each group, it was higher than the initial weight. During the experiment, the group treated with CTX gained more weight than the other four groups, although this difference was not statistically significant (*p* > 0.05). Within three days of CTX exposure, there was no significant difference in body weight between the CTX group and the sample-treated group; however, following 5 days of CTX exposure, the difference in weight gain became apparent [24]. Similarly, three days following the CTX injection, there was no significant difference between groups in terms of weight gain in this study. The results suggested that intragastric administration of CBH and HD did not affect the growth in immunosuppressed mice.

The immune organ indexes can reflect the immunity moderation level to a certain extent. CTX is a widely used drug in chemotherapy, displaying a significant anti-cancer ability. As an inducer commonly used in immunosuppression models, CTX can damage the immune defense of the host and suppress immune organs, immune cells, and immune molecules [25]. As shown in Figure 2A,B, the spleen and thymus indexes in the CTX group were significantly lower (*p* < 0.05) than in the Normal group, indicating that intraperitoneal injection of CTX resulted in immune organ atrophy, successfully establishing the immunosuppression model. As previously reported, CTX can significantly increase the apoptotic rate of spleen and thymus cells [26]. Furthermore, levamisole, CBH, and HD significantly increased the spleen and thymus indexes (*p* < 0.05) compared to the Model group, while no significant differences were apparent among the three groups. Levamisole was initially developed as an anthelmintic. As an immunomodulator, the drug garnered considerable attention later on. According to the reports, levamisole has numerous immunomodulatory effects, including the enhancement of antibody production to various antigens, the enhancement of several cellular immune responses, synergistic activity with T-lymphocyte mitogens, the enhancement of chemotaxis and the enhancement of the phagocytic activity of polymorphonuclear and mononuclear phagocytes [27,28]. Similar to levamisole, the results demonstrated that CBH and HD supplements could prevent and treat the immune organ atrophy caused by CTX. Yu et al. [29] found that peptides from *Nibea japonica* skin could significantly improve the spleen and thymus indexes of CTX-induced immunosuppressed mice. Banerjee et al. [30] revealed that CAR treatment prevented spleen atrophy caused by immunosuppression. Liu et al. [31] intragastrically administered phenylalanine dipeptide to mice, significantly improving the spleen and thymus indexes of the mice. Based on the results mentioned above, CBH effectively improved the atrophy of immune organs caused by CTX. Considering that this has the same effect as HD supplementation, it can be inferred that HD is the primary substance in CBH, preventing the immune organ atrophy caused by CTX. Therefore, low molecular peptides, such as dipeptides, display immunomodulatory activity to prevent immune organ atrophy.

### 3.3. The Effect of CBH on the Proliferation of Splenic LYM

The spleen is a unique organ that combines innate and adaptive immune systems. The spleen is important for immunoregulation as well as playing a part in the immunological response [32]. The spleen is the place where mature LYM settle, with B cells comprising around 60% of the total amount of spleen LYM and T cells comprising around 40%. The spleen receives a higher volume of LYM on a daily basis than all other secondary lymphoid organs combined [33]. Consequently, these splenocytes could represent in vivo systemic immune responses. LYM proliferation is an important event during the activation process of the adaptive immune system. T LYM is primarily responsible for regulating the cellular immunity of the body. The most striking feature of T LYM activation is the development of mitotic proliferation. T LYM immunity is frequently detected using Con A-induced LYM proliferation [34].

In this study, splenocytes isolated from all the groups were cultured with Con A, and their proliferation is presented in Figure 2C. With Con A treatment, the LYM proliferative ratio was lower in the CTX group (*p* < 0.05), whereas the splenocyte proliferation in CBH and HD groups, was higher than in the CTX group (142.22 and 150.98%, respectively). Moreover, there were no discernible changes among the Normal, Levamisole, CBH, and HD groups. These results indicated that through enhancing cellular immune systems, CBH and HD could promote Con A-induced LYM proliferation. Since the same concentrations and proportions of HD were present in the CBH and HD groups, HD may represent the primary functional components in CBH, preventing the decrease in splenic LYM proliferation caused by CTX. Furthermore, some investigations have found that the food supply influences splenocyte growth. Many low molecular peptides can enhance the proliferative ability of splenic LYM and display immunomodulatory activity. Yang et al. [35] showed that the peptides in *Pseudostellaria heterophylla* protein hydrolysate could restore splenic LYM proliferation in mice treated with CTX. Hou et al. [36] isolated three immunomodulatory peptides (including two pentapeptides and one dipeptide) from the Alaska pollock frame enzymatic hydrolysates, which significantly increased the proliferation rate of splenic LYM in mice. The results of this study indicated that CBH protects cellular immunity during CTX-induced immune damage.

### 3.4. The Effect of CBH on the Hematological Parameters

WBCs, such as NEU and LYM, represent crucial immune cells, the number of which directly reflects the humoral immunity of the body [37]. NEUs are the most common leukocytes in peripheral blood, accounting for 50–70% of all the circulating leukocytes in humans and 20% of all circulating leukocytes in mice. NEU performs a vital phagocytic function and represents the initial line of defense against microbial pathogen invasion. LYM, such as T and B LYM and natural killer cells, make up roughly 20–30% of all the circulating leukocytes in humans and are mainly involved in the specific immune response of the body [38]. Previous studies have shown that CTX caused protein functional groups to be alkylated, inhibiting medulla hemopoietic functionality and reducing the number of WBCs in the blood [39].

Table 1 demonstrates that CTX reduced the WBC, NEU, LYM, and PLT counts in the blood (*p* < 0.05), while increasing the RBC and HGB content (*p* < 0.05), showing that the immunity of the mice was suppressed. One of the most noticeable side effects of chemotherapy drugs is bone marrow suppression. WBC are believed to be involved in immunological and defensive mechanisms, have a short lifetime, and require bone marrow stem cells to develop continuously [40]. The second day of CTX treatment deeply affected bone marrow architecture, and recovery began on day 5. Thus, there was a dramatic drop in white blood cell count in the blood of mice 3 days after the CTX injection. Juaristi et al. [41] suggest that the proliferation and differentiation of erythroid progenitor cells after the acute early injury inflicted by CTX, is associated with changes in EPO-R expression during spontaneous recovery. Therefore, the increased RBC and HGB levels in the CTX group may be related to EPO-R expression promotion. This study is consistent with previous studies that CTX significantly reduced the amount of WBC and PLT in the blood while increasing the content of RBC and hemoglobin HGB [42]. Compared to the CTX group, CBH prevented a decrease in WBC, NEU, LYM, and PLT (*p* < 0.05), alleviating the RBC and HGB (*p* < 0.05) increase in the blood. Consequently, the immunosuppression caused by CTX was improved. The HD group had considerably greater WBC, LYM, and PLT counts than the CTX group (*p* < 0.05). RBC and HGB levels, on the other hand, were considerably lower in the HD group than in the CTX group (*p* < 0.05). These results suggest that CBH and HD prevented and treated CTX-induced WBCs decline, with an effect comparable to levamisole. Therefore, HD may represent the main functional components in CBH that prevent the decrease in WBCs caused by CTX. Previous studies have shown that immunoregulatory selenium-enriched peptides from soybean can restore the CTX-induced decrease of WBCs [43]. Lis et al. [44] revealed that a low molecular weight dipeptide bestatin significantly improved the CTX-induced decrease in the number of peripheral blood LYM. These results indicated that the administration of CBH and HD restored normal blood indices, implying that CBH and HD protect against immunosuppression induced by CTX.

### 3.5. The Effect of CBH on the Immunoglobulin Levels

Serum immunoglobulins are critical markers of humoral immunity and are involved in immune response and regulation [45]. Secreted IgM, IgG, and IgA represent the primary antibody components in serum and are vital effector molecules during the humoral immune response. During the early stages of the first humoral immune response, IgM is the primary antibody produced. It is the largest molecular weight immunoglobulin with a strong bactericidal, bacteriolytic, hemolytic, phagocytotic, and anti-infection ability [45]. In the phagocytotic process of monocytes, IgG, the most abundant and prominent antibody in serum, plays a crucial role [46]. IgG is more likely to diffuse through the capillary wall to the interstitial space, playing an essential anti-infection, toxin neutralization, and conditioning role. IgA, being the primary antibody class in the secretions that bathe these mucosal surfaces, serves as an important first-line of defense. IgA acts against various microbial antigens and can neutralize toxins and viruses [47].

To examine the effect of CBH and HD on the humoral immunity of CTX-treated mice, the serum IgM, IgG, and IgA levels of each group were determined. The results are shown in Figure 3A–C, respectively. CTX significantly reduced serum IgM, IgG, and IgA levels as compared to the Normal group (*p* < 0.05). Compared with the CTX group, the IgM and IgA levels were substantially higher in the serum of the Levamisole, CBH, and HD groups (*p* < 0.05), while no significant differences were evident between these three groups. Furthermore, no significant differences were apparent between the serum IgA levels of the CBH and Normal groups (*p* > 0.05). This indicated that CBH effectively prevented a decrease in the serum IgA content caused by CTX, maintaining it at a normal level. Although the IgG levels in the Levamisole, CBH, and HD groups exceeded that in the CTX group, the differences were not significant (*p* > 0.05). The results showed that the supplementation of CBH or HD prevented a CTX-induced decrease in the immunoglobulin content and that the effect was similar to that of levamisole. Both pretreatment and treatment mediated the effect of CBH and HD on increasing immunoglobulin content in this study. The findings also suggest that HD may represent the main functional components in CBH responsible for preventing a CTX-induced decrease in serum immunoglobulin.

B LYM primarily participates in the humoral immune response and is transformed into plasma cells via antigen stimulation. Major humoral immune components include immunoglobulins, which are secreted by plasma cells (differentiated B cells). Immunoglobulins interact with immune response mediators and specific cell receptors to mediate a variety of protective activities [26]. Peptides stimulate the receptors on the surface of the B cells, causing them to proliferate and differentiate, transforming them into plasma cells to produce immunoglobulin [48]. Yu et al. [49] investigated the in vivo immunomodulatory activity of a novel pentadecapeptide RVAPEEHPVEGRYLV (SCSP) from a cellular and humoral immunity perspective in a CTX-induced immunosuppression mouse model. The results indicated that SCSP significantly enhanced the serum IgA, IgM, and IgG levels of the mice. In this study, HD may also increase the number and activity of B cells by acting on the receptors on the B cell surfaces, increasing the level of serum immunoglobulin.

### 3.6. The Effect of CBH on the Cytokine Levels and Gene Expression

Cytokines are synthesized and secreted by immune cells (B cells, T cells, and NK cells) and non-immune cells (endothelial cells, epidermal cells, osteoblasts, neurons, and fibroblasts), which can regulate immune functions [50,51]. Furthermore, cytokines play a crucial function in the intercellular communication of the immune system. They regulate the maturation, proliferation, and responsiveness of certain cell populations, as well as the balance between humoral and cell-based immune responses. The secretion of cytokines is a crucial indicator of immune function in the body. IL-2 has been shown to improve T cell killing activity, cause T cells to secrete IFN-γ, increase NK cell differentiation and activation, and promote B LYM proliferation and differentiation as well as immunoglobulin synthesis [52]. IFN-γ is a key element for the immune system to effectively act against infections and is produced mostly by NK cells, Th1, and CTL. IFN-γ can activate macrophages, enhance their phagocytotic ability, improve NK cell activity, increase antibacterial and anti-tumor ability, and stimulate antigen presentation [53]. Negative regulators are also important for immune response maintenance. IL-10 is a key Th2 negative regulatory cytokine. IL-10 plays a crucial immunosuppressive and anti-apoptotic role by suppressing the release of inflammatory mononuclear macrophage factors. In addition, IL-10 may inhibit the activation of inflammatory cytokines (such as IL-1, IL-12, and TNF-α) and chemokines (MCP family) secreted by antigen-presenting cells, indirectly inhibiting the function of T cells [54]. Cytokines are often used as indicators for evaluating immune response regulation in experiments. IL-2 can promote the activation and proliferation of T cells and NK cells, resulting in the secretion of IFN-γ and the enhancement of immune function [55]. IL-10 can reflect immunosuppression levels. TNF-α, which is secreted primarily by mononuclear macrophages, is an essential component of the host defense system and can induce the expression of other immunoregulatory and inflammatory mediators to eliminate the tumor cell. TNF-α is not secreted primarily by splenic lymphocytes; therefore, TNF-α levels were not measured. In this study, IL-2, IL-10, and IFN-γ were measured in the supernatants of splenic lymphocyte cultures to investigate the regulatory effect of CBH and HD on immune suppression.

It is generally acknowledged that splenic LYM represents the principal immune response effector cells, synthesizing various immunomodulation factors, including cytokines and cell adhesion molecules. To further illustrate the immunomodulatory action of CBH and HD, the levels of IL-2, IFN-γ, and IL-10 in the splenic LYM were measured. As shown in Figure 4A,B, the levels of IL-2 and IFN-γ in the splenic LYM of the CTX group were markedly reduced compared with the Normal group (*p* < 0.05), while these reductions were observed to a lesser extent in the Levamisole, CBH, and HD groups (*p* < 0.05). Figure 5C demonstrates that the levels of IL-10 in the splenic LYM of the CTX group were considerably higher than the Normal group (*p* < 0.05). However, the concentration of IL-10 in the Levamisole, CBH, and HD groups was markedly lower than in the CTX group (*p* < 0.05). In addition, the effect of CBH on the downregulation of IL-10 level exceeded that of HD, and no differences were evident between the CBH and Normal groups. Although this may be attributed to the contribution of other functional peptides in the CBH to the downregulation of IL-10, HD provided 72% of its down-regulation ability. These findings indicated that CBH improved the immune responses by increasing IL-2 and IFN-γ secretion while decreasing IL-10 secretion in the splenic LYM of the CTX-induced immunosuppressed mice. Moreover, HD represents the main functional CBH component for regulating cytokine secretion. Jia et al. [56] found that peptides extracted from calf spleens can improve the immune function of CTX-induced immunosuppressed mice by regulating the levels of cytokines, such as IL-2, IL-10, and IFN-γ. Xu et al. [57] revealed that Gly-Gln dipeptide could significantly increase the IL-2 content secreted by the blood and splenic LYM of mice.

To further confirm the effect of CBH and HD on the regulation of the three cytokines secreted by the splenic LYM, the induction of the IL-2, IFN-γ, and IL-10 transcriptional regulation was investigated using qRT-PCR. As shown in Figure 4D,E, the mRNA expression of IL-2 and IFN-γ were substantially lower in the splenic LYM of the CTX group than in the Normal group (*p* < 0.05), while these reductions were observed to a lesser extent in the Levamisole, CBH, and HD groups (*p* < 0.05). However, the mRNA expression of IFN-γ was substantially higher in the CBH group than in the HD group. Figure 4E illustrates that the IL-10 mRNA expression in the splenic LYM of the CTX group was considerably higher than in the Normal group (*p* < 0.05). However, the mRNA expression of IL-10 in the Levamisole, CBH, and HD groups was markedly lower than in the CTX group (*p* < 0.05). IL-10 mRNA expression was substantially lower in the CBH group than in the HD group. The results showed that CBH and HD enhanced the IL-2 and IFN-γ mRNA expression while inhibiting IL-10 mRNA expression in the splenic LYM. Therefore, the IL-2 and IFN-γ secretion was increased, while that of IL-10 was reduced. Yoo et al. [58] found that *Phellinus baumii* extract contained HD that increased the IFN-γ mRNA expression and other cytokines in the spleen of immunosuppressed mice induced by CTX. In this study, CBH and HD played an immunomodulatory role by regulating the mRNA expression of IL-2, IFN-γ, and IL-10 in the splenic LYM, regulating the content of the three cytokines secreted by the splenic LYM. It is speculated that HD represents the functional component in CBH with significant immunomodulatory ability. In this study, CBH and HD mediated their regular effects on cytokine secretion from splenic LYM through a combination of pretreatment before CTX injection and therapeutic effects after CTX injection.

### 3.7. The Effect of CBH on the NO/cGMP and cAMP/PKA Signaling Pathways

When external stimuli act on the receptors on LYM surfaces, second messengers are activated via transmembrane transmission. The second messenger transmits the signal to the downstream protein kinase, where it becomes phosphorylated to regulate gene expression and cell function. cAMP and cGMP are the primary second messengers in cells catalyzed by adenylate cyclase (AC) and guanylate cyclase (GC), respectively [59].

During the immune response, cGMP is involved in cell differentiation, chemotaxis, cell proliferation, and the release of soluble mediators. NO plays a crucial role in inflammation and immunity, with numerous physiologic and pathophysiologic effects [60]. NO is generated from the L-arginine and catalyzed by one of three NO synthase (NOS) isoforms: neuronal NOS (nNOS), endothelial NOS (eNOS), and inducible NOS (iNOS) [61]. The only isoform of NOS really implicated in the immune response is the iNOS, while nNOS and eNOS are constitutively activated [21,62]. The primary physiological stimulant of cGMP synthesis is NO, which activates soluble GC (sGC). In addition, activation of the sGC pathway results in the production of cGMP. Once produced, cGMP exerts its function through protein kinase G (PKG).

This study determined the NO and cGMP content in the splenic LYM of the mice. Figure 5A,B show that the NO and cGMP levels in the CTX group were significantly lower (*p* < 0.05) than in the Normal group. The NO and cGMP levels in the Levamisole, CBH, and HD groups were considerably higher than in the CTX group (*p* < 0.05), and no significant differences (*p* > 0.05) were apparent between the three groups. The results showed that the CBH and HD could significantly increase the NO and cGMP content in the splenic LYM of immunosuppressed mice. The activation of the NO/cGMP pathway can promote cell proliferation [63]. Carvalho et al. [64] showed that decreased cGMP and NO levels in T LYM decreased the cytokines mRNA levels, such as IFN-γ and IL-2. In this study, CBH and HD regulated the proliferation of splenic LYM via the NO/cGMP signaling pathway and regulated cytokine gene transcription, affecting cytokine secretion and fulfilling an immunomodulatory function.

Signal transduction via the cAMP/PKA pathway is essential for many processes in a variety of cells. cAMP is a crucial component in the G protein-mediated signaling pathway. When exposed to external stimuli, the intracellular cAMP content can rapidly increase over a short time, forming signal molecules. PKA is made up of tetramers consisting of two regulatory and two catalytic subunits that are found in every cell and govern a variety of functions. When two cAMP molecules are bound by the regulatory subunits, the catalytic subunits are released, phosphorylating their target proteins.

This study determined the cAMP and PKA activity in the splenic LYM of the mice. Figure 5C,D show that the cGMP levels and PKA activity in the CTX group were significantly lower (*p* < 0.05) than in the Normal group. The cGMP levels and PKA activity in the Levamisole, CBH, and HD groups were substantially higher than in the CTX group (*p* < 0.05). The results showed that CBH and HD could significantly increase the cAMP content and PKA activity in the splenic LYM of immunosuppressed mice. Liopeta et al. [65] demonstrated the suppressive effect of cAMP/PKA on the IL-10 levels. Zuo et al. [66] found that an increase in the cAMP and PKA levels in rat spleen cells reduced the serum IL-10 levels. In this study, the CBH and HD may regulate the transcription and secretion of cytokines via the cAMP/PKA signaling pathway. The activation of the NO/cGMP and cAMP/PKA intracellular signaling cascades in immune cells suggests that pretreatment and treatment with CBH and HD display immunomodulatory activity.

## 4. Conclusions

In summary, this study provides in vivo evidence that CBH and HD have an immunomodulatory effect on CTX-induced immunosuppression in mice. CBH and HD significantly increase the spleen and thymus indexes. Moreover, peripheral WBC, RBC, HGB, and PLT show that CBH and HD treatment inhibits CTX-induced immunosuppression in mice, restoring splenocyte proliferation. Furthermore, IgM, IgG, and IgA secretion are modified following CBH and HD treatment, significantly increasing IL-2 and IFN-γ production and mRNA expression in splenic LYM. CBH and HD also significantly decrease IL-10 production and mRNA expression in splenic LYM. Moreover, CBH and HD play an immunomodulatory role in splenic LYM via the cAMP/PKA and NO/cGMP signaling pathways. The immunotherapy effects of CBH and HD in this study included both a preventive effect from 0 to 12 days and a therapeutic effect after CTX injection. Although, in theory, many molecules can potentially be responsible for the immunomodulatory effect, this study suggests that HD plays a pivotal role in CBH. These observations indicate that CBH supplements exhibit considerable potential for preventing immune system suppression.

## Figures and Tables

**Figure 1 nutrients-14-04491-f001:**
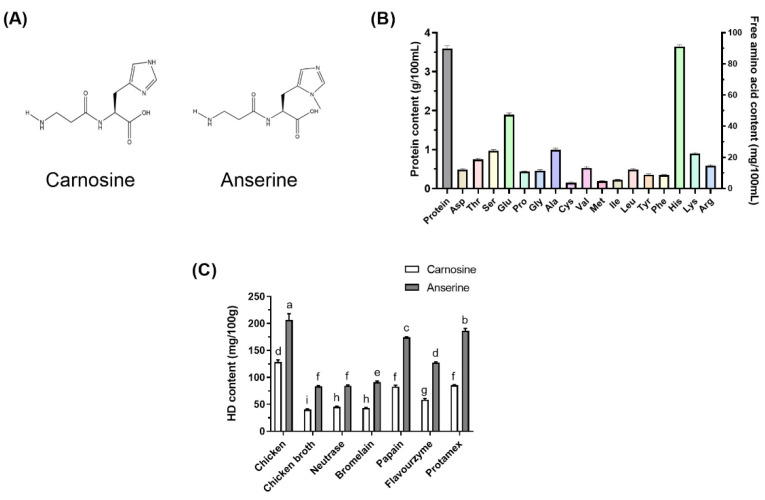
The structure of CAR and ANS (**A**). The protein and free amino acid content in CB (**B**). The effect of different enzymatic hydrolytic treatments on the HD content in the CB (**C**). Different letters indicated significant differences among the groups (*p* < 0.05).

**Figure 2 nutrients-14-04491-f002:**
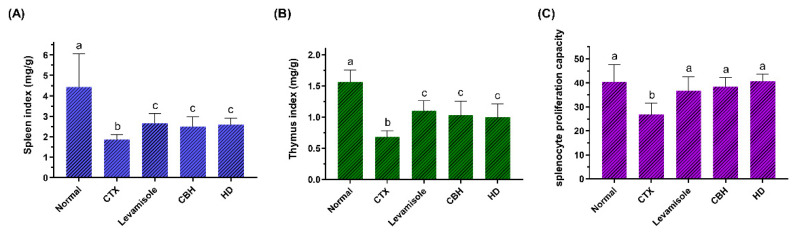
The effect of CHB and HD on the spleen index (**A**), thymus index (**B**), and the proliferation of splenocytes stimulated with Con A (7.5 µg/mL) (**C**). Data are presented as means ± SD (*n* = 10). Different letters for the same index among the groups represent significant differences at *p* < 0.05.

**Figure 3 nutrients-14-04491-f003:**
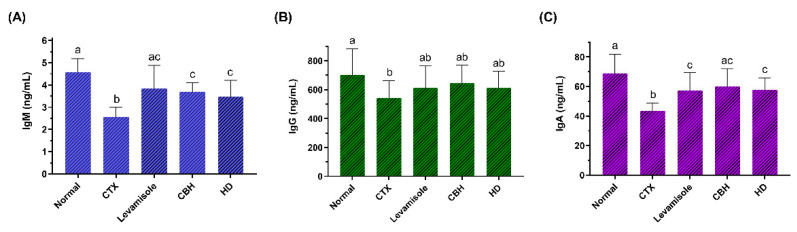
The effect of CHB and HD on the IgM (**A**), IgG (**B**), and IgA (**C**) content in the serum of CTX-treated mice. Data are presented as means ± SD (*n* = 10). Different letters for the same index among the groups represent significant differences at *p* < 0.05.

**Figure 4 nutrients-14-04491-f004:**
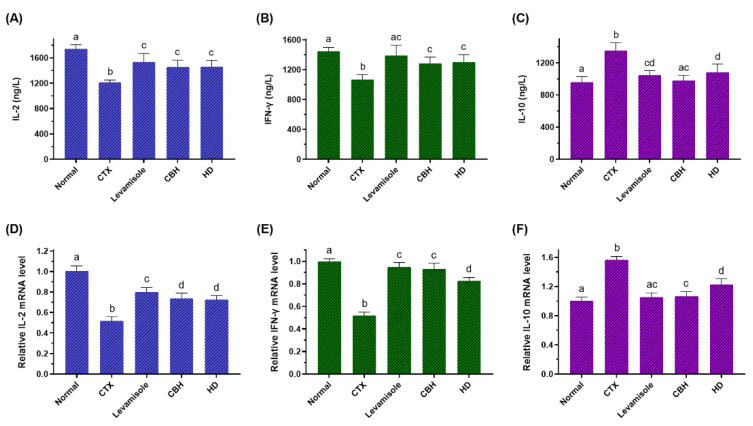
The effect of CHB and HD on the IL-2 (**A**), IFN-γ (**B**), and IL-10 (**C**) levels in the splenocytes, and the mRNA expression levels of IL-2 (**D**), IFN-γ (**E**), and IL-10 (**F**) in the spleens of CTX-treated mice. Data are presented as means ± SD (*n* = 10). Different letters for the same index among the groups represent significant differences at *p* < 0.05.

**Figure 5 nutrients-14-04491-f005:**
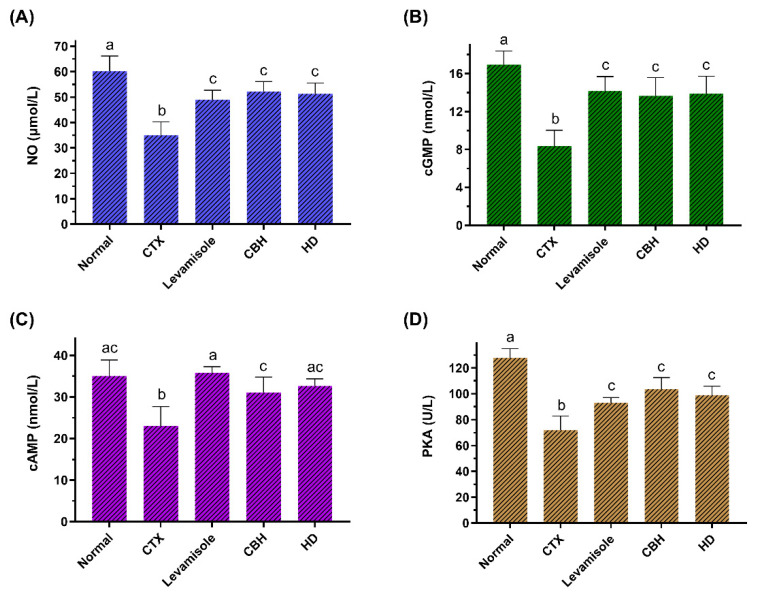
The effect of CHB and HD on the NO (**A**), cGMP (**B**), and cAMP (**C**) content, as well as the PKA (**D**) activity in the splenocytes of CTX-treated mice. Data are presented as means ± SD (*n* = 10). Different letters for the same index among the groups represent significant differences at *p* < 0.05.

**Table 1 nutrients-14-04491-t001:** The effect of CHB and HD on the body weight and hematological parameters of CTX-treated mice.

	Normal	CTX	Levamisole	CBH	HD
Initial weight (g)	36.30 ± 0.78 ^a^	34.57 ± 1.35 ^b^	36.04 ± 1.21 ^a^	35.86 ± 1.43 ^ab^	35.85 ± 1.84 ^ab^
Final weight (g)	38.49 ± 1.31 ^a^	36.91 ± 1.85 ^a^	37.69 ± 1.51 ^a^	36.87 ± 2.31 ^a^	37.16 ± 2.45 ^a^
Weight gain (g)	2.19 ± 0.99 ^a^	2.34 ± 0.90 ^a^	1.65 ± 1.02 ^a^	1.01 ± 1.75 ^a^	1.31 ± 1.04 ^a^
WBC (K/μL)	7.72 ± 2.80 ^a^	1.46 ± 0.67 ^b^	2.50 ± 0.46 ^c^	3.04 ± 0.56 ^c^	2.70 ± 0.43 ^c^
NEU (K/μL)	2.01 ± 1.13 ^a^	0.50 ± 0.34 ^b^	0.80 ± 0.17 ^bc^	1.08 ± 0.34 ^ac^	0.76 ± 0.17 ^bc^
LYM (K/μL)	5.21 ± 1.92 ^a^	0.79 ± 0.34 ^b^	1.32 ± 0.27 ^c^	1.80 ± 0.46 ^c^	1.65 ± 0.36 ^c^
RBC (M/μL)	10.18 ± 1.76 ^ac^	11.69 ± 1.80 ^b^	11.22 ± 1.14 ^ab^	10.28 ± 0.54 ^ac^	9.68 ± 1.07 ^c^
HGB (g/dL)	16.20 ± 2.97 ^ac^	18.66 ± 2.52 ^b^	17.58 ± 1.74 ^ab^	16.29 ± 0.80 ^ac^	15.19 ± 1.42 ^c^
PLT (K/μL)	1104.90 ± 166.68 ^a^	712.10 ± 223.60 ^b^	855.50 ± 235.86 ^bc^	939.60 ± 215.84 ^ac^	1029.22 ± 154.29 ^ac^

Values are expressed as means ± SD (*n* = 10). WBC, white blood cell count; NEU, neutrophils; LYM, lymphocytes; RBC, red blood cell count; HGB, hemoglobin concentration; PLT, platelet count. Means in the same row with different letters differ significantly (*p* < 0.05).

## Data Availability

The data that support the findings of this study are available from the corresponding author upon reasonable request.

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
