# Peer review of "Immunomodulatory Effects of Chicken Broth and Histidine Dipeptides on the Cyclophosphamide-Induced Immunosuppression Mouse Model"

_nutrients, 2022, doi:10.3390/nu14214491_

Round 1

Reviewer 1 Report

The manuscript is convincing in demonstration that ingredients of the chicken broth are effective in reconstitution of several immune parameters. However, it will require some corrections and explanations.

Major

Although experimental groups were randomly completed (n=10) the initial mouse weight in the groups could be more equal for the sake of data interpretation. Why the initial weight of CP- treated mice was lower? The weight gain in CP-treated group was 2.34 g versus 2.19 g in the control. It looks that CP did not make any harm to the mice despite the significant weight loss in lymphoid organs. Maybe it results from a beneficial effect of pretreatment with the immune stimulants.

The authors should clearly state that the observed beneficial effects of the broth peptides on CP-induced changes are caused both by the pretreatment (preventive effect), days 0-12,  and the treatment (days 13-15). Although the resultant observations are encouraging the experiment lacks a main control such as analysis of immune parameters on day 12, before CP treatment. It is obvious that treatment of normal mice with the investigated preparations would also positively change the studied parameters.  This would, however, require usage of a double number of mice. Anyway, this limitation should be commented.

The degree of sensitivity to CP of cells is very different. Neutrophils are most sensitive because of their extremely short half life, B cells are also strongly affected regarding their long -term ability to repopulate lymphoid organs. T-cell dependent immune response is quickly restored. The nadir in neutrophil content occurs on day 4 after single CP injection so the protocol applied by the authors ensures more or less a good suppressive effect. It must be however kept in mind, that a rapid rebound in neutrophil level occurs on day 8 with about 80% of neutrophils in circulation. Similar changes regard erythropoiesis (elevated erythrocytes and hemoglobin), so may be the enclosed article will help the authors with interpretation of this phenomenon.

Julián Antonio Juaristi , María Victoria Aguirre, Juan Santiago Todaro, Mirta Alba Alvarez, Nora Cristina Brandan. EPO receptor, Bax and Bcl-x(L) expressions in murine erythropoiesis after cyclophosphamide treatment. Toxicology. 2007 Mar 7;231(2-3):188-99

Minor

These statements should be corrected

This is  true that neutrophils represent 50-70% of blood leukocytes, but only in humans. In rodents, only about 20% of circulating leukocytes are neutrophils, including  their precursors, since they are predominantly stored in bone marrow. In case of infection they are rapidly exported into circulation to reach about 40% content of blood leukocytes.

This is not true that splenocytes contain only 10% of antigen presenting cells (maybe if we take into account all cell types, including erythrocytes). Spleen contains more B cells than T cells and all B cells are perfect APC since they all express MHC class II antigens essential for antigen presentation. In addition, what about splenic macrophages?

Figure 4. descriptions of Y axis should be reversed.

Author Response

Reviewer 1:

  1. The reviewer's comment: Although experimental groups were randomly completed (n=10) the initial mouse weight in the groups could be more equal for the sake of data interpretation. Why the initial weight of CP-treated mice was lower? The weight gain in CP-treated group was 2.34 g versus 2.19 g in the control. It looks that CP did not make any harm to the mice despite the significant weight loss in lymphoid organs. Maybe it results from a beneficial effect of pretreatment with the immune stimulants.

The authors' answer: We thank the reviewer for the comment. In this study, after a week of acclimatization, the mice were randomly assigned to five groups based on body weight using the randomized design method. Randomization resulted in a lower initial body weight of CTX-treated mice. To observe the effect of CHB and HD treatment on weight changes in immunosuppressed mice, we added data on the weight gain of the mice in the manuscript (page 6, line 298). During the experiment, the weight of mice in each group increased, and the weight gain in CTX-treated group was higher than that in the Control group, but the difference in weight gain of each group was not significant. It has been reported that within three days of CTX exposure, there was no significant difference in body weight between the CTX group and the drug-treated group, and the difference in weight gain was manifested after 5 days of CTX exposure. Likewise, three days after CTX injection, there was no significant difference in the weight gain of each group in this study. The results indicate that there is no general toxicity by oral administration of CHB and HD. We have revised the text to address your concerns and hope that it is now clearer. Please see page 6 of the revised manuscript, lines 289-295.

References:

  1. Yu, F.; Zhang, Z.; Ye, S.; Hong, X.; Jin, H.; Huang, F.; Yang, Z.; Tang, Y.; Chen, Y.; Ding, G. Immunoenhancement effects of pentadecapeptide derived from Cyclina sinensis on immune-deficient mice induced by Cyclophosphamide. Funct. Foods 2019, 60, 103408, doi:10.1016/j.jff.2019.06.010.
  2. Yu, F.; He, K.; Dong, X.; Zhang, Z.; Wang, F.; Tang, Y.; Chen, Y.; Ding, G. Immunomodulatory activity of low molecular-weight peptides from Nibea japonica skin in cyclophosphamide-induced immunosuppressed mice. Funct. Foods 2020, 68, 103888, doi:10.1016/j.jff.2020.103888.
  3. Jiang, X.; Yang, F.; Zhao, Q.; Tian, D.; Tang, Y. Protective effects of pentadecapeptide derived from Cyclaina sinensis against cyclophosphamide-induced hepatotoxicity. Biophys. Res. Commun. 2019, 520, 392-398, doi:10.1016/j.bbrc.2019.10.051.

  1. The reviewer's comment: The authors should clearly state that the observed beneficial effects of the broth peptides on CP-induced changes are caused both by the pretreatment (preventive effect), days 0-12, and the treatment (days 13-15). Although the resultant observations are encouraging the experiment lacks a main control such as analysis of immune parameters on day 12, before CP treatment. It is obvious that treatment of normal mice with the investigated preparations would also positively change the studied parameters. This would, however, require usage of a double number of mice. Anyway, this limitation should be commented.

The authors' answer: We agree with the reviewer that the observed beneficial effects of the CBH and HD on CTX-induced changes are caused both by the pretreatment on days 0-12 and the treatment on days 13-15 in this study. As stated by the reviewer, treatment of normal mice with CBH and HD may also be effective in altering immunosuppressive parameters. This study suggests that HD plays a key role in the immunomodulatory of CBH. The literature confirmed that carnosine pretreatment reduced the genotoxicity induced by CTX in bone marrow cells of mice. Furthermore, Xu et al. showed that carnosine administration for continuous seven days could substantially improve suppressed hematopoietic functions induced by CTX. This indicates that HD has both preventive and therapeutic effects on immunosuppression. This study did not measure the immune markers on days 0-12 before CP injection. In this study, after 12 days of CBH and HD pretreatment, mice were intraperitoneal injected with CTX for 3 days, and CBH and HD was administered by gavage at the same time. Therefore, we have supplemented the limitations of our experimental results in the conclusion section of the manuscript (page 13, line 632).

References:

  1. Naghshvar, F.; Abianeh, S.M.; Ahmadashrafi, S.; Hosseinimehr, S.J. Chemoprotective effects of carnosine against genotoxicity induced by cyclophosphamide in mice bone marrow cells. Cell Biochem. Funct. 2012, 30, 569-573, doi:10.1002/cbf.2834.
  2. Xu, M.; He, R.-R.; Zhai, Y.-J.; Abe, K.; Kurihara, H. Effects of Carnosine on Cyclophosphamide-Induced Hematopoietic Suppression in Mice. The American Journal of Chinese Medicine 2014, 42, 131-142, doi:10.1142/s0192415x14500098.

  1. The reviewer's comment: The degree of sensitivity to CP of cells is very different. Neutrophils are most sensitive because of their extremely short half life, B cells are also strongly affected regarding their long-term ability to repopulate lymphoid organs. T-cell dependent immune response is quickly restored. The nadir in neutrophil content occurs on day 4 after single CP injection so the protocol applied by the authors ensures more or less a good suppressive effect. It must be however kept in mind, that a rapid rebound in neutrophil level occurs on day 8 with about 80% of neutrophils in circulation. Similar changes regard erythropoiesis (elevated erythrocytes and hemoglobin), so may be the enclosed article will help the authors with interpretation of this phenomenon.

The authors' answer: We appreciate the reviewer’s insightful suggestion. The leukocyte changes induced by CTX in this experiment are in accordance with the reviewer's description. In future experimental design, we will pay attention to the reviewer's reminder about the situation that the leukocyte level decreased first and then increased after CP injection. In the present study, RBC and HGB levels increased in the CTX group 3 days after CTX injection compared with the Normal group. We carefully read the articles recommended by the reviewers. It was mentioned that CTX extremely affected erythropoiesis, with the most apoptosis on the second day and the recovery on the fourth day. And the EPO-R expression seemed to be crucial for the precise regulation of survival, proliferation and differentiation of the erythroid cells. The increased RBC and HGB levels in this study may be related to the promotion of EPO-R expression. And we have revised the text to address your concerns in the revised manuscript (page 8, lines 399-409). “The lifetime of WBC is short and needs bone marrow stem cells to differentiate continuously complement [40]. The whole architecture of bone marrow was deeply affected on the second day and the recovery towards normality began from day 5 following CTX treatment. Thus, 3 days after CTX injection, the number of white blood cells in the blood of mice was significantly reduced. Juaristi et al. [41] suggest that the proliferation and differentiation of erythroid progenitor cells after the acute early injury inflicted by CTX, is associated with changes in EPO-R expression during spontaneous recovery. Therefore, the increased RBC and HGB levels in the CTX group may be related to the promotion of EPO-R expression. Studies showed that CTX significantly reduced the amount of WBC and PLT in the blood, and increased the content of RBC and hemoglobin HGB, which is consistent with this study [42].”

References:

  1. Juaristi, J.A.; Aguirre, M.V.; Todaro, J.S.; Alvarez, M.A.; Brandan, N.C. EPO receptor, Bax and Bcl-x(L) expressions in murine erythropoiesis after cyclophosphamide treatment. Toxicology 2007, 231, 188-199, doi:10.1016/j.tox.2006.12.004.
  2. Arslan, S.; Ozyurek, E.; Gunduz-Demir, C. A color and shape based algorithm for segmentation of white blood cells in peripheral blood and bone marrow images. Cytometry Part A 2014, 85A, 480-490, doi:10.1002/cyto.a.22457.
  3. Chu, Q.; Zhang, Y.; Chen, W.; Jia, R.; Yu, X.; Wang, Y.; Li, Y.; Liu, Y.; Ye, X.; Yu, L., et al. Apios americana Medik flowers polysaccharide (AFP) alleviate Cyclophosphamide-induced immunosuppression in ICR mice. J. Biol. Macromol. 2020, 144, 829-836, doi:10.1016/j.ijbiomac.2019.10.035.

  1. The reviewer's comment: This is true that neutrophils represent 50-70% of blood leukocytes, but only in humans. In rodents, only about 20% of circulating leukocytes are neutrophils, including their precursors, since they are predominantly stored in bone marrow. In case of infection they are rapidly exported into circulation to reach about 40% content of blood leukocytes.

The authors' answer: We thank the reviewer for pointing this out. We have made the change in the revised manuscript (page 8, line 388). The new sentence is revised as follows. “NEUs are the most common WBCs in peripheral blood, accounting for 50%-70% of all the circulating leukocytes in humans and accounting about 20% of circulating leukocytes in mice.”

  1. The reviewer's comment: This is not true that splenocytes contain only 10% of antigen presenting cells (maybe if we take into account all cell types, including erythrocytes). Spleen contains more B cells than T cells and all B cells are perfect APC since they all express MHC class II antigens essential for antigen presentation. In addition, what about splenic macrophages?

The authors' answer: We agree with the reviewer and have revised the text as follows (page 7, line 347). “The spleen is the place where mature LYM settle, among which B cells account for about 60% of the total number of spleen LYM, and T cells account for about 40%.” In addition, the spleen contains about a quarter of the body’s total lymphocyte population; during lymphocyte recirculation, more cells pass through the spleen than through all the lymph nodes. In this experiment, only splenic lymphocytes were isolated, but splenic macrophages were not collected. Therefore, in this study, the proliferation ability of spleen LYM was measured to analyze the immunomodulatory effect of CBH and HD, and the splenic macrophages were not measured. In future studies, we will consider adding a macrophage phagocytic capacity assay.

  1. The reviewer's comment: Figure 4. descriptions of Y axis should be reversed.

The authors' answer: We are grateful for the suggestion and apologize the errors in the Figure 4. We have made corrections to the ordinal labels of the figures (page 11, line 539).

We acknowledge and thank the reviewers for the comments and suggestions, which are valuable in improving the quality of our manuscript. All the modified indicated above appear in the revised manuscript.

We would also like to thank you for allowing us to resubmit a revised copy of the manuscript, which we hope will be accepted for publication in the Nutrients.

Thank you to all the reviewers for the kind advice.

Reviewer 2 Report

Zhang and coauthors in this manuscript describe the immunomodulatory effects of chicken broth dipeptides. The experimental work seems well done, however there are some critical points that deserve to be improved.

Majors:

a)       Abstract should be improved describing for example the route of administration of CBH and HD and adding some numeric results and some p values to the description of the results.

b)      In the introduction authors stated that chicken broth contains high levels of PUFA. In European chickens the overall level of PUFA is lower than that of saturated and monounsaturated ones. If this data is different for the Hetian chickens used, then the reference in which it is demonstrated should be added.

c)       CB preparations should be nutritionally characterized, at least for total protein content and fats (saturated, mono and polyunsaturated FA). Otherwise, it would be impossible to replicate the experiments with chickens of different breed, origin or type of farming.

d)      In the same way the CBH and HD dosage should be characterized at least for total amino acids or fats orally administrated to mice.

Minor

The rationale for Levamisole (anthelmintic) use should be better explained.

The choice to dose IL-2, IL-10 and IFNg and not for example TNFa must be explained. The rationale for the choice of these cytokines and not of other ones is missing.

Table 1: the name “Model” in the column should be changed with CTX group.

Author Response

Reviewer 2:

  1. The reviewer's comment: Abstract should be improved describing for example the route of administration of CBH and HD and adding some numeric results and some p values to the description of the results.

The authors' answer: Thank you for the suggestion. We have supplemented the route of administration of CBH and HD and added p values to the description of the results in the abstract.

  1. The reviewer's comment: In the introduction authors stated that chicken broth contains high levels of PUFA. In European chickens the overall level of PUFA is lower than that of saturated and monounsaturated ones. If this data is different for the Hetian chickens used, then the reference in which it is demonstrated should be added.

The authors' answer: We agree with the reviewer that in European chickens the overall level of PUFA is lower than that of saturated and monounsaturated ones. A study reported that the PUFA level in Hetian chicken is about 23%, the SFA level is about 48%, and the MUFA level is about 29%. Although PUFA levels in Hetian chickens were lower than SFA and MUFA, it still accounts for a high proportion. The CB in this study was boiled and filtered to remove most of the fat. The total fat content of the CB was 0.6%, and the total fat content of the CBH was 1.1%. We considered that the low residual fat content in CB did not affect the functionality of the HD, therefore, did not present fat data in the paper. Meanwhile, the immunomodulatory effect of CBH in this study was mainly provided by HD, not PUFA. To avoid misunderstandings, we have removed PUFA and added relevant references in the Introduction section of the revised manuscript (page 1, line 34).

References:

  1. Yang, Y.; Feng, Y.; Li, Z.; Xie, X.; Miao, F.; Fang, G.; Tu, H.; Wen, J. Effect of sex and diet nutrition on the contents of flavor precursors in fujian hetian chicken. Acta Vet. Zootech. Sin. 2006, 37, 242-249, doi:10.3321/j.issn:0366-6964.2006.03.006.

  1. The reviewer's comment: CB preparations should be nutritionally characterized, at least for total protein content and fats (saturated, mono and polyunsaturated FA). Otherwise, it would be impossible to replicate the experiments with chickens of different breed, origin or type of farming.

The authors' answer: We thank the reviewer for pointing this out. In this study, the total protein content of CB was 3.59 g/100mL, and the total fat content was 0.6%. Hetian chicken was purchased from the Hetian Chicken Development Co., Ltd. (Fujian, China). The ingredients used to prepare the CB were sampled from eight breeding lots of the aforementioned company. The reproducibility of the experiment was ensured by mixing stable commercial multi-batch ingredients in the preparation of CB. HD is present in substantial levels in chicken and fulfill many physiological functions in the human body. The purpose of this study was to investigate the immunomodulatory effects of CBH and HD on immunosuppressed mice. HD may be a critical functional component of CBH that plays an immunoregulatory role. For this reason, we determined HD-related protein content and free amino acid content in CB and supplemented in Figure 1 (B) (page 2, line 55), Section 2.3 (page 4, line 169) and Section 3.1 (page 5, line 262).

Figure 1. The structure of CAR and ANS (A). The protein content and free amino acid content in CB (B). The effect of different enzymatic hydrolytic treatments on the HD content in the CB (C). Different letters indicated significant differences among the groups (P<0.05).

  1. The reviewer's comment: In the same way the CBH and HD dosage should be characterized at least for total amino acids or fats orally administrated to mice.

The authors' answer: We agree with the reviewer that further determination of total amino acids or fats to characterize CBH and HD would be helpful. The CB in this study was boiled and filtered to remove most of the fat. The total fat content of the CB was 0.6%, and the total fat content of the CBH was 1.1%. We considered that the low residual fat content in CB did not affect the functionality of the HD, therefore, did not present fat data in the paper. In revised manuscript, we have supplemented the determination of protein content and free amino acid content in CB at Figure 1 (B) (page 2, line 55) and Section 3.1 (page 5, line 262). At the same time, this study investigated the ability of CBH and HD to alleviate immunosuppression based on the high content of HD in CB and the immune activity. Therefore, we determined the CAR and ANS content in chicken breast, CB, and CBH. The results showed that enzymatic hydrolysis treatments significantly increased the HD content in the CB. Finally, CBH and HD were administered orally to mice to determine the immunological activity of CBH and the critical role of HD in CBH. Therefore, in this study, we used CAR and ANS content, to unify the dose of CBH and HD by gavage to ensure the reproducibility of the experiment. The relevant data are described on page 4, line195 of the revised manuscript. The HD concentration in the CBH was 30 mg/mL, including 9.41 mg/mL CAR and 20.59 mg/mL ANS. The HD doses were equivalent to the HD concentration and proportion in the CBH.

  1. The reviewer's comment: The rationale for Levamisole (anthelmintic) use should be better explained.

The authors' answer: We thank the reviewer for the comment. Levamisole, a drug that had been originally designed for antihelminthic applications. Later, the drug received considerable attention as an immunomodulator. Levamisole is used to boost immunity in a number of human diseases as well, including leprosy, some cancers. Levamisole has been reported to have a broad range of immunomodulatory effects, including the enhancement of antibody production to several different antigens, augmentation of a variety of cellular immune responses (such as an increase in lymphokine production by lymphocytes), synergistic activity with T-lymphocyte mitogens, augmentation of chemotaxis, and enhancement of the phagocytic activity of polymorphonuclear leukocytes and mononuclear phagocytes. Currently, clinical commonly used immunopotentiating agents include chemosynthetic drugs like levamisole and isopropyinosine. Therefore, levamisole was used as a positive drug for improving immunity in this study. At the same time, we have supplemented the rationale for levamisole use in the revised manuscript. (page 7, lines 322-328)

References:

  1. Stevenson, H.C.; Green, I.; Hamilton, J.M.; Calabro, B.A.; Parkinson, D.R. Levamisole-known effects on the immune-system, clinical-results, and future applications to the treatment of cancer. Clin. Oncol. 1991, 9, 2052-2066, doi:10.1200/jco.1991.9.11.2052.
  2. Sajid, M.S.; Iqbal, Z.; Muhammad, G.; Iqbal, M.U. Immunomodulatory effect of various anti-parasitics: a review. Parasitology 2006, 132, 301-313, doi:10.1017/s0031182005009108.
  3. Chu, Q.; Zhang, Y.; Chen, W.; Jia, R.; Yu, X.; Wang, Y.; Li, Y.; Liu, Y.; Ye, X.; Yu, L., et al. Apios americana Medik flowers polysaccharide (AFP) alleviate Cyclophosphamide-induced immunosuppression in ICR mice. Int. J. Biol. Macromol. 2020, 144, 829-836, doi:10.1016/j.ijbiomac.2019.10.035.

  1. The reviewer's comment: The choice to dose IL-2, IL-10 and IFN-γ and not for example TNF-α must be explained. The rationale for the choice of these cytokines and not of other ones is missing.

The authors' answer: We are grateful for the suggestion. We have supplemented the rationale for the choice of IL-2, IL-10 and IFN-γ in the revised manuscript (page 10, line 506). In this study, we chose to measure the levels of IL-2, IL-10 and IFN-γ in splenic lymphocyte culture supernatants. In the manuscript, it has been proved that CBH and HD can promote the proliferation of splenic lymphocytes, and significantly promote the proliferation of T lymphocytes induced by ConA. Earlier reports also indicated that carnosine supplementation could inhibit lymphocyte apoptosis, stimulate lymphocyte proliferation and regulate immune function, such as extending the lifespan of CD4+ T cells, which leads to the enhancement of IL-2, IFN-γ and IL-12 productions, while suppressing IL-10 production. IL-2, also known as T cell growth factor, is widely used to promote the activation and proliferation of T cells and NK cells. It plays an important role in promoting the proliferation of spleen lymphocytes. IFN-γ is produced by activated T cells and NK cells, which can mediate cellular immune function and promote the differentiation and proliferation of Th1 cells. IL-2, IFN-γ and IL-10 are mainly produced by lymphocytes, while TNF-α is mainly produced by mononuclear macrophages. In this experiment, the content of cytokines in the supernatant of splenic lymphocytes was measured, so the contents of IL-2, IFN-γ and IL-10 were determined, but TNF-α was not selected.

References:

  1. Zhang, W.; Gong, L.; Liu, Y.; Zhou, Z.; Wan, C.; Xu, J.; Wu, Q.; Chen, L.; Lu, Y.; Chen, Y. Immunoenhancement effect of crude polysaccharides of Helvella leucopus on cyclophosphamide-induced immunosuppressive mice. Funct. Foods 2020, 69, 103942, doi:10.1016/j.jff.2020.103942.
  2. Li, Y.; He, R.; Tsoi, B.; Li, X.; Li, W.; Abe, K.; Kurihara, H. Anti-Stress Effects of Carnosine on Restraint-Evoked Immunocompromise in Mice through Spleen Lymphocyte Number Maintenance. PLoS One 2012, 7, e33190, doi:10.1371/journal.pone.0033190.
  3. Hyland, P.; Duggan, O.; Hipkiss, A.; Barnett, C.; Barnett, Y. The effects of carnosine on oxidative DNA damage levels and in vitro lifespan in human peripheral blood derived CD4+T cell clones. Mechanisms of Ageing and Development 2000, 121, 203-215.

  1. The reviewer's comment: Table 1: the name “Model” in the column should be changed with CTX group.

The authors' answer: Thank you for the suggestion. We have modified the name “model” to “CTX” in Table 1. (page 6, line 298)

We acknowledge and thank the reviewers for the comments and suggestions, which are valuable in improving the quality of our manuscript. All the modified indicated above appear in the revised manuscript.

We would also like to thank you for allowing us to resubmit a revised copy of the manuscript, which we hope will be accepted for publication in the Nutrients.

Thank you to all the reviewers for the kind advice.

Sincerely yours,

He Li and Jian Zhang